# Molecular Profiling and the Interaction of Somatic Mutations with Transcriptomic Profiles in Non-Melanoma Skin Cancer (NMSC) in a Population Exposed to Arsenic

**DOI:** 10.3390/cells13121056

**Published:** 2024-06-18

**Authors:** Farzana Jasmine, Maria Argos, Yuliia Khamkevych, Tariqul Islam, Muhammad Rakibuz-Zaman, Mohammad Shahriar, Christopher R. Shea, Habibul Ahsan, Muhammad G. Kibriya

**Affiliations:** 1Institute for Population and Precision Health (IPPH), University of Chicago, Chicago, IL 60637, USA; farzana@uchicago.edu (F.J.);; 2Epidemiology & Biostatistics, Global Health, University of Illinois Chicago, Chicago, IL 60612, USA; 3UChicago Research Bangladesh (URB), University of Chicago, Dhaka 1230, Bangladesh; 4Pulse Infoframe, London, ON N5X 4E7, Canada; 5Division of Dermatology, Department of Medicine, University of Chicago, Chicago, IL 60637, USA; 6Department of Public Health Sciences, Biological Science Division, University of Chicago, Chicago, IL 60637, USA

**Keywords:** somatic mutation, basal cell carcinoma, squamous cell carcinoma, arsenic exposure, hedgehog signaling, notch signaling, iL-17 signaling, gene–environment interaction

## Abstract

Exposure to inorganic arsenic (As) is recognized as a risk factor for non-melanoma skin cancer (NMSC). We followed up with 7000 adults for 6 years who were exposed to As. During follow-up, 2.2% of the males and 1.3% of the females developed basal cell carcinoma (BCC), while 0.4% of the male and 0.2% of the female participants developed squamous cell carcinoma (SCC). Using a panel of more than 400 cancer-related genes, we detected somatic mutations (SMs) in the first 32 NMSC samples (BCC = 26 and SCC = 6) by comparing paired (tissue–blood) samples from the same individual and then comparing them to the SM in healthy skin tissue from 16 participants. We identified (a) a list of NMSC-associated SMs, (b) SMs present in both NMSC and healthy skin, and (c) SMs found only in healthy skin. We also demonstrate that the presence of non-synonymous SMs in the top mutated genes (like *PTCH1*, *NOTCH1*, *SYNE1*, *PKHD1* in BCC and *TP53* in SCC) significantly affects the magnitude of differential expressions of major genes and gene pathways (basal cell carcinoma pathways, *NOTCH* signaling, *IL-17* signaling, *p53* signaling, Wnt signaling pathway). These findings may help select groups of patients for targeted therapy, like hedgehog signaling inhibitors, IL17 inhibitors, etc., in the future.

## 1. Introduction

The human epidermis is composed of keratinocytes, which exhibit cuboidal (basaloid) cytology at the lowest (basal) layer and squamous cytology at the suprabasal layers. It also contains Merkel cells and pigment-producing melanocytes. Tumors originating from keratinocytes or Merkel cells are grouped into non-melanoma skin cancer (NMSC). NMSC includes basal cell carcinoma (BCC) and squamous cell carcinoma (SCC). Bowen disease (BD) is a clinical term for squamous cell carcinoma in situ. In 2019, there were 4.0 million cases of BCC and 2.4 million cases of SCC worldwide [1]. In 2012, in the United States (US), 56,987 patients were identified with BCC (39,035 incident and 17,952 prevalent) [2]. NMSC is the most prevalent malignancy in the US, exceeding all other cancers combined with an estimated 2 million new diagnoses each year [3,4]. Both BCC and SCC have low mortality but can have a high recurrence rate and can cause significant disfiguration, particularly in the head and neck regions where they commonly occur [4,5]. BCC of skin is the most common type and may account for about 90% of all skin cancers [6,7,8].

The incidence of BCC shows a strong inverse correlation with geographic latitude combined with the pigment status of its inhabitants [9]. The highest rates are seen in Australia, where over one in two inhabitants will be diagnosed with BCC by the time they reach 70 years of age [10]. The incidence rates in Asia and South America are ten- to hundred-fold lower [11,12,13]. Patients with a BCC have a seventeen-fold increased risk of subsequent BCC compared with the general population, as well as a three-fold increased risk of subsequent SCC and a two-fold increased risk of melanoma [14,15]. The mortality of BCC is extremely low. However, the healthcare cost for NMSC is quite high. A US Medicare expenditure study showed that NMSC was the fifth most costly cancer between 1992 and 1995 [16]. A report estimated the average annual cost of treating NMSC in the US to be USD 4.8 billion from 2007 to 2011, a substantial increase compared with the 2002 to 2006 estimate of USD 2.7 billion [17].

The risk factors for both BCC and SCC include ultraviolet radiation (UVR) and chronic immunosuppression [18,19,20]. UVR is the major known environmental risk factor. Thus, the prevention strategy is photoprotection, which can be both topical and systemic [21]. The ability to repair UV-induced DNA damage reduces with age. Increasing age and the male sex (at older age) are well-known factors for an increased risk of BCC. Molecular biomarkers of NMSC including genomics, transcriptomics, proteomics, and metabolomics have been recently reviewed extensively [22,23]. Arsenic (As) is a known carcinogen that appears in groundwater and is associated with skin cancer [24]. Chronic exposure to As may induce BCC [25,26,27,28]. One study suggested that miR-155-5p regulates the NF-AT1-mediated immunological dysfunction that is involved in the pathogenesis and carcinogenesis of As [29]. Some studies showed that in the presence of As exposure, decreased telomere length predisposes individuals to an increased risk of BCC [30]. As generates reactive oxygen species that cause oxidative stress, leading to DNA damage. Concurrently, As inhibits DNA repair, modifies the epigenetic regulation of gene expression, and targets protein function due to its ability to replace zinc in select proteins [31]. In a recent study, we have shown that high As exposure was associated with impaired DNA replication pathways, cellular response to different DNA damage repair mechanisms, and immune response [32].

In Bangladesh, between 2000 and 2002, 11,746 participants (5042 men and 6704 women) were recruited for the Health Effects of Arsenic Longitudinal Study (HEALS) and were exposed to As through the consumption of As-enriched groundwater. The study found 714 confirmed cases of premalignant skin lesions [33].

UVR-induced cancers, such as BCC and melanoma, exhibit the highest prevalence of somatic mutations. The majority show UVR signatures. Two tumor suppressor genes are important in sporadic BCC: patched 1 (*PTCH1*) and tumor protein 53 (*TP53*). Loss of heterozygosity in chromosome 9q22 is the most frequently encountered cytogenic change in BCC. The inactivation of *PTCH1* and the up-regulation of hedgehog (Hh) signaling are most likely pivotal events in the pathogenesis [34]. Studies have shown that a large percentage (~85%) of BCC may harbor somatic mutations in Hh pathway genes—*PTCH1* (73%), *SMO* (20%), and *SUFU* (8%) [3,35]. Mutations in *TP53* are also found in >60% of cases [3,35,36]. In BCC, mutations are also reported in other known cancer genes, including *NRAS*, *HRAS*, *KRAS*, *PIK3CA*, *RAC1*, *FBXW7*, *RB1*, *CDKN2A*, *NOTCH1*, *NOTCH2*, *CASP8*, and *ARID1A* [3,36]. Other significantly mutated genes include *PTN14*, *MYCN*, *RPL22*, and *PPIAL4G* [3]. There is evidence that Wnt pathways play a role in the pathogenesis of NMSCs through the activation of inflammatory processes and the stimulation of cancer cell proliferation and invasion [37]. A recent study highlighted the role of *CYFIP2*, *HOXB5*, *PTPN3*, *MARCKSL1*, *PTCH1*, and *CDC2* in BCC [38].

Our group conducted a double-blind, placebo-controlled study, the Bangladesh Vitamin E and Selenium Trial (BEST), to evaluate the effect of vitamin E and selenium supplementation in the prevention of NMSC in a population exposed to As who had a clinical manifestation of As toxicity in the form of As-related non-malignant skin lesion [39]. Seven thousand subjects were followed up for 6 years for the development of NMSC. Of them, 1.7% developed BCC (males = 2.2%, females = 1.3%) [32]. Our previous study showed interactions of As exposure and gene expression profiling in BCC in the study group who were exposed to inorganic As through drinking As-contaminated well water [32].

The mutational profile in SCC tissue may be slightly different [40,41,42,43]. In a study on high-risk head and neck SCC patients, *TP53* was mutated in 100% of cases; *APC*, *ATM*, *ERBB4*, *GNAQ*, *KIT*, *RB1*, and *ALB1* were altered in 60% of cases; and *FGF2* mutation was seen in 40% of cases [43]. Interestingly, druggable targets may be found in 60–80% of the cases [40,43].

Gene expression studies [34,44,45,46,47] (mostly with small sample sizes) in NMSC have suggested differential expressions of Wnt and Hh pathway genes [34], TGF-beta signaling, PPAR-Υ signaling pathway genes [48], and MAPK pathways [47] for classical BCC. SCC-like BCC may show differential expressions of immune-response genes and oxidative stress-related genes [34]. There are few transcriptome-wide gene expression studies in SCC [45,49,50,51]. Compared to normal skin, SCC and actinic keratosis show differential expression of a number of genes, including *RAB31*, *MAP4K4*, *IL-1RN*, *NMI*, and *IL4R*; however, there was no difference between SCC and actinic keratosis [45].

Studies on NMSC have been mainly performed in Caucasian populations. To our knowledge, no study addresses the molecular profiling of NMSC in a “non-Caucasian population” exposed to As. Moreover, no study in NMSC has yet addressed the fact that sunlight exposure is associated with a large number of somatic mutations in different genes in non-lesional, apparently healthy skin. Therefore, identifying a somatic mutation in NMSC samples does not necessarily establish the association between that mutation and the NMSC pathogenesis. In this study, we have looked for (a) somatic mutations in NMSC tissue by scanning more than 400 cancer-related genes to identify NMSC-associated somatic mutations in a Bangladeshi population exposed to As through drinking As-contaminated water and (b) examined if such mutation(s) were associated with differential expression of gene(s) or pathway(s).

## 2. Materials and Methods

### 2.1. Study Population

For this study, we selected the first 32 subjects from the BEST study developing histopathologically confirmed NMSC and had the tumor tissue properly preserved in RNA later, an RNA stabilizing buffer (ThermoFisher Scientific, Waltham, MA, USA). The BEST study included 7000 men and women (m = 2840, f = 4160) who were known to be exposed to As through consuming well water containing As [32,39]. This study included all subjects with clinically visible non-malignant skin lesions (melanosis, leukomelanosis, or keratosis)—a known manifestation of As toxicity. We also collected “non-lesional” or apparently healthy skin tissue surrounding the margin of the arsenical keratosis lesion from 16 independent patients and preserved it in the same way. Throughout the manuscript, we have used the term “healthy skin tissue” for these non-lesional skin tissues. Patient characteristics are shown in Appendix A. All these patients were followed up every 2 years for a total of 6 years to check for the development of NMSC. The urinary As creatinine ratio (UACR) was also recorded at baseline and follow-up visits. A skin biopsy was performed on patients who had reasonable clinical suspicion of BD or NMSC, including SCC and BCC. All these patients consented to a biopsy. During this follow-up period, 14.7% of the males and 7.5% of the females (*p* < 0.0001, chi-square test) had a skin biopsy performed. Histopathological examination was performed by two pathologists independently. For the pathological diagnosis, a structured reporting form was used (see Appendix A). A skin biopsy was performed on a total of 727 participants (m = 417, f = 310). Among them, 37.7% of the biopsies showed BD, 2.9% had invasive SCC, 16.1% showed BCC, and the rest (43.3%) showed arsenical keratosis or other skin conditions [32]. Thus, among the As-exposed study population, 2.2% of the male and 1.3% of the female participants developed BCC, while 0.4% of the male and 0.2% of the female participants developed SCC over the six-year follow-up.

### 2.2. Arsenic Exposure Measurement

We measured UACR at baseline and 2-year, 4-year, and 6-year follow-up as a measure of As exposure. The urinary total As concentration was measured by inductively coupled plasma mass spectrometry [52]. Urinary creatinine was measured by a colorimetric method based on the Jaffe reaction described by Heinegard and Tiderstrom [53]. The urinary As was measured from a spot urine collection. To take into account the hydration status, we used the UACR as a measure of As exposure. The log_2_-transformed UACR showed a strong correlation to the log_2_-transformed well water as a concentration (r = 0.66) [54].

### 2.3. Nucleic Acid Extraction

DNA was extracted from these RNA later preserved tissues using a Quick-DNA/RNA Microprep Plus kit (Zymo Research, Irvine, CA, USA) following the manufacturer’s protocol. After taking the samples out of the RNA later, the tissue was washed with a 1xPBS buffer and then submerged into the DNA/RNA shield before simultaneous DNA and RNA extraction. RNA and DNA quantification and the 260/280 ratio were checked by NanoDrop 1000.

### 2.4. Somatic Mutation Assay

For the detection of somatic mutation, we used AmpliSeq for the Illumina comprehensive Cancer Panel Guide (Illumina Inc., San Diego, CA, USA). The total gene list is presented in Appendix A. There were 4 plates with 4 different sets of primer pools. We made 4 identical plates of DNA with 10 ng input DNA each for 1 pool plate. DNA target regions were amplified in all 4 plates, and then amplified DNA was pooled together in corresponding wells on one plate. In the next step, primer dimer or unused amplicon were digested. After that, i7 and i5 adapters were ligated to the amplicon ends. These products were cleaned up by magnetic beads and then amplified for the 2nd time. The library was then cleaned up to have the final library for sequencing. After pooling, the final library was measured with a fluorometer. Library size and quantity were also measured by a fragment analyzer. Sequencing was performed on the Illumina HiSeq platform (San Diego, CA, USA).

### 2.5. Gene Expression Assay

For RNA sequencing on the Illumina platform, we used Lexogen’s QuantSeq 3′ mRNA–Seq kit (Vienna, Austria) for library preparation as described previously [32]. The final library was measured by a fluorometer, and after pooling, qPCR was performed to quantify the input library for sequencing on the Illumina HiSeq platform (San Diego, CA, USA).

This study was approved by the Institutional Review Board of The University of Chicago Medicine protocol code IRB19-0724 and was approved on 24 September 2019.

### 2.6. Statistical Method

Mutation detection: The FASTQ Illumina sequencing data were initially processed by CLC genomics Workbench23 (https://digitalinsights.qiagen.com/ (accessed on 26 April 2023)). After adapter sequencing trimming, default parameters were used for QC. The minimum length was kept at 40. A Targeted Amplicon Sequencing (TAS) module for paired samples was used where we used the tissue sample and the corresponding blood DNA sample as a pair. In this module, initially, the reads were mapped to homo sapiens sequence hg19, and the variants were detected using structural variant caller v1.2 (Biomedical Genomics Analysis 23.1). Variants found in normal samples (in our case—the blood) were removed from the variants detected in the tissue sample. The in-built workflow removed the germline variants found in the public database (db SNP, 1000 genomes project, dbSNPs common, and hapmap) that were found in the mapped reads. Also, variants outside the target region were removed as they are likely to be false positives due to non-specific mapping of sequencing reads. The parameters for the low-frequency variant detection were set at a minimum coverage of 10, a minimum count of 2, and a minimum frequency of 2%. Next, the remaining variants (the “somatic variants”) were annotated with gene names, amino acid changes, conservation scores, and information from ClinVar (variants with clinically relevant association). We used a variant calling quality score of Q60 as the cut-off for the list of somatic mutations.

Transcriptome data were processed using Partek Flow (version 10.0) (https://www.partek.com/partek-flow/, accessed on 11 November 2022). A STAR aligner was used for alignment, and the final gene count data were expressed as the count per million reads (CPM) and were log_2_ transformed for the ANOVA using Partek Genomics Suite (version 7.0) (https://www.partek.com/partek-genomics-suite/, accessed on 22 April 2024). For statistical analyses, IBM SPSS Statistics version 29 was used. We also used Partek Genomis Suit for ANOVA and Gene set ANOVA as described in a previous paper [32]. In the GO enrichment analysis, we tested if the differentially expressed genes (as per the set criteria) fell into a Gene Ontology category more often than expected by chance. We used a chi-square test for comparison. The negative log of the *p*-value for this test was used as the enrichment score. In addition to GO enrichment analysis, we also examined the differential expression of “gene sets” using the Gene Set Enrichment Analysis (GSEA) [55]. Given an a priori-defined set of genes “S” (sharing the same GO category or the KEGG pathway), the goal of GSEA was to determine whether the members of “S” were randomly distributed throughout the ranked list or primarily found at the top or bottom. For further statistical comparison of the magnitudes of the differential expression of the “Gene set” in the absence or presence of a factor (mutation), we used “Gene set ANOVA”, which offers an introduction to the interaction terms in the model. Gene set ANOVA is a mixed model ANOVA to test the expression of a set of genes (sharing the same category or functional group) instead of an individual gene in different groups. The analysis is performed at the gene level, but the result is expressed at the level of the Gene set category by averaging the member genes’ results. The equation for the model is as follows:Model: Y = μ + T + G + TxG + TxMut + ε
where Y represents the expression status of a Gene set category, μ is the common effect or average expression of the Gene set category, T is the tissue-to-tissue (tumor/normal) effect, G is the gene-to-gene effect, TxG is the differential pattern of gene expression in different tissue types, TxMut is the interaction term, and ε represents the random error.

## 3. Results

The diagnoses of BCC (n = 26) and SCC (n = 6) were confirmed by skin biopsy. There was consensus between two pathologists for all 32 cases.

### 3.1. Somatic Mutation

Considering the fact that even healthy-looking skin tissue is also exposed to sunlight and may develop UVR-induced somatic mutations, for each tissue sample, we compared the tissue DNA with the corresponding whole blood DNA (a proxy for germline) from the same patient for the detection of a somatic mutation. In 32 tumor tissue samples, we found a total of 6829 somatic mutations (in 3385 unique genomic loci, see Figure 1A). In 16 healthy skin tissues, we found a total of 2530 somatic mutations in 1470 unique genomic loci (see Figure 1A). Some of the variant metrics are shown in Table 1.

The median number of somatic mutations per BCC sample was 148; for SCC, it was 180.5 per sample; and for healthy skin tissue, it was 140 per sample (*p* = 0.73, Kruskal–Wallis test). Considering the target sequence region of 1.7 Mb, the calculated median tumor mutation burden (TMB) was 87 mutations/Mb for BCC tissue, 106 mutations/Mb for SCC tissue, and 82 mutations/Mb for healthy skin tissue (*p* = 0.58, Kruskal–Wallis test). When we compared the TMB in BCC, SCC, and healthy skin tissue by sex, the difference was not statistically different, although the TMB appeared to be higher in females.

We generated a list of somatic mutations in NMSC cases (BCC and SCC) and a list of somatic mutations in healthy skin tissue from an independent set of participants. Then, by comparing the somatic mutations in tumor tissue and healthy skin tissue, we looked for (a) NMSC-associated somatic mutations, (b) somatic mutations potentially associated with NMSC, which are found in tumor tissue as well as in healthy skin tissue, and (c) somatic mutations present only in healthy skin tissue.

The overlap of the total unique somatic mutation loci between tumor tissue and healthy skin tissue and the mutations stratified by type (SNV, Del, and INS) are shown in Figure 1A. Among the SNVs, irrespective of BCC, SCC, or healthy skin tissue, the most common type of substitution was C > T (median 15.7% of substitutions/sample, 95% CI 1.6–37.8%) and G > A (median 15.8% of substitutions/sample, 95% CI 0–42.7%) without statistical difference between the tissue types (Appendix A). This high prevalence of C > T + G > A substitution is consistent with the mutational signature for NMSC usually related to sunlight exposure.

### 3.2. NMSC-Associated Somatic SNVs

There were a total of 1611 somatic SNVs (representing 1440 unique SNV loci in 361 genes) detected only in NMSC samples (total n = 32, of which BCC = 26, SCC = 6) and not in healthy skin tissue (Figure 1B). All of these were in the gene coding regions; 321 were found in the ClinVar database, 628 were also found in TCGA skin cancer samples and reported in the COSMIC database, and 604 were “non-synonymous” SNVs. Among these 1611 NMSC-associated somatic SNVs, 1344 SNVs (covering 1222 loci in 344 genes) were found in BCC and the other 267 SNVs (covering 261 loci in 153 genes) were found in SCC. Some 43 unique loci were common in BCC and SCC but not in normal skin tissue. The list of the top 20 genes harboring these BCC-associated and SCC-associated somatic mutations are shown in Figure 2A and Figure 2B, respectively.

### 3.3. Somatic Mutation SNVs Common in NMSC and Healthy Skin Tissue

We detected 277 somatic SNVs (representing 139 unique SNV loci in 95 different genes) found in both NMSC and healthy skin (Figure 1B). A total of 96 of them were reported in ClinVar; 66 were also found in TCGA skin cancer samples and reported in the COSMIC database; and 34 were “non-synonymous” SNVs. The list of the top twenty genes harboring these somatic mutations potentially associated with NMSC is shown in Figure 3A.

### 3.4. Somatic Mutation SNVs Detected Only in Healthy Skin Tissue

We detected 426 somatic SNVs (representing 401 unique SNV loci in 192 different genes), which were found only in healthy skin and not in any NMSC tissue (see Figure 1B). A total of 87 of them were reported in ClinVar; 146 were also found in TCGA skin cancer samples and reported in the COSMIC database; and 83 were “non-synonymous” SNVs. The list of the top twenty genes harboring these somatic mutations potentially associated with NMSC is shown in Figure 3B.

### 3.5. Association of Somatic Mutation and Differential Gene Expression in NMSC

In the next step, we asked if the absence or presence of somatic mutation(s) in the tissue showed a difference in differential gene expression patterns in tumor tissue compared to healthy skin tissue. Considering the fact that non-synonymous SNVs (causing amino acid change) may have functional effects, we restricted the analysis to NMSC-associated non-synonymous SNVs only. So, a tumor tissue was only considered a mutant for *PTCH1* (for example) if that sample harbored at least one of the non-synonymous SNVs in *PTCH1* but not if it harbored only some other SNVs in the *PTCH1* gene.

#### 3.5.1. Gene Level Analysis

A comparison of gene expression data between BCC (n = 26) and healthy skin tissue (n = 16) showed that 118 genes were differentially expressed at least by a fold change (FC) of 3 and an FDR level of ≤0.05 (see Appendix A). Gene Ontology (GO) or enrichment analysis of this gene list is shown in Figure 4. The list was enriched in genes involved in “Basal cell carcinoma”, “Hedgehog signaling pathway”, and “pathways in cancer”. It may be mentioned that GSEA analysis (see Appendix A) also confirmed the enrichment of these pathway genes.

Next, in the ANOVA model(s), we entered an interaction term “tissue (0 = healthy, 1 = BCC) x nonsynonymous mutation in *PTCH1* (0 = no mutation, 1 = mutation)” to find out the genes that had a different magnitude of differential expression in the BCC tissue in the absence or presence of the mutation. The differential expression of these same 118 differentially expressed genes in the absence (n = 14) and the presence of the non-synonymous somatic mutation (n = 12) in the *PTCH1* gene compared to the same normal skin (n = 16) are presented in Table 2 along with the interaction *p*-values. In the combined analysis, the *PTCH1* gene was overexpressed in BCC tissue by FC = 4 (95% CI 2.2–7.2) compared to healthy skin tissue (see Appendix A); but, in the absence of any non-synonymous somatic mutation in *PTCH1* in BCC tissue (n = 14), the FC was 2.8 (95% CI 1.5–5.3), and in presence of a non-synonymous somatic mutation in *PTCH1* in the BCC tissue (n = 12), the FC was 6 (95% CI 3.1–11.8) (interaction *p* = 0.03, see Table 2). The result showed that the magnitude of differential expression for 40 out of these 118 genes was statistically different if the tumor had the non-synonymous mutation in *PTCH1* (see the interaction *p* column in Table 2). In fact, the effect of this somatic mutation in *PTCH1* was more pronounced for other genes.

Similarly, we asked whether the absence or presence of somatic mutations in *NOTCH1* (Appendix A), *SYNE1* (Appendix A), *PKHD1* (Appendix A), and *EP400* (Appendix A) in BCC tissue was associated with a difference in magnitude of the differential expression of genes. The result shows that the non-synonymous somatic mutations of each of these genes have a significant association with functional effects in terms of differential gene expression. In the next step, we wanted to see the effect on the gene pathway level.

#### 3.5.2. Pathway Level Analysis

In this step, we examined if a set of genes (e.g., in the KEGG pathway) was differentially expressed in NMSC tissue compared to normal skin tissue and if the magnitude of differential expression in NMSC compared to normal was significantly different in the absence or presence of non-synonymous somatic mutations in tumor tissue. First, we looked at the mutation of the *PTCH1* gene. Table 3 shows the BCC-associated non-synonymous somatic mutations in the *PTCH1* gene found only in tumor tissue but not in healthy skin tissue.

In a previous study, using the gene expression data from the same 26 BCC samples, we have shown that the top differential pathways in BCC include the “hedgehog signaling pathway” and the “basal cell carcinoma pathway”, and there was an interaction with the degree of As exposure in this population [32].


**Association of the Somatic ns Mutation in the PTCH1 Gene and Dysregulated Pathways in BCC**


The detailed results from Gene set ANOVA for all the KEGG pathways are presented in Appendix A. Compared to healthy skin tissue, in the BCC samples without the *PTCH1* non-synonymous somatic mutation (n = 14), the genes in the “Basal cell carcinoma pathway” were overexpressed by FC 1.62 (95% CI 1.34–1.96), whereas in BCC samples with the *PTCH1* non-synonymous somatic mutation (n = 12), the same pathway genes were overexpressed by FC 3.95 (95% CI 3.24–4.82). This shows a significant association (interaction *p* = 2.48 × 10^−17^) between *PTCH1* mutation status and the overexpression of the “basal cell carcinoma pathway”. Among the other major pathways that are markedly overexpressed in the presence of the *PTCH1* mutation include the “hedgehog signaling pathway” and the “TGF-beta signaling pathway” (see Figure 5).

We also conducted the rank-based analysis—GSEA for patients without the *PTCH1* ns mutation (Appendix A) and for patients with the *PTCH1* ns mutation (Appendix A). It was interesting to note that in the GSEA analysis, too, many of the pathways found in the above-mentioned Gene set ANOVA were also seen to be more significantly enriched in the presence of the *PTCH1* mutation.


**Association of the Somatic ns Mutation in the NOTCH1 Gene and Dysregulated Pathways in BCC**


In the same way, we looked at the *NOTCH1* mutation status (nineteen BCC without and seven BCC with the *NOTCH1* mutation) and compared it to the same healthy skin tissue (n = 16). The major pathways that were more markedly overexpressed in the presence of *NOTCH1* mutation include the “IL-17 signaling pathway”, “peroxisome related genes”, and “NF-Kappa beta signaling pathway”, and the “TGF beta signaling pathway” (see Figure 6).


**Association of the Somatic ns Mutation in SYNE1 and PKHD1 Genes and Dysregulated Pathways in BCC**


For BCC, we also looked for associations with other frequently mutated genes, such as *SYNE1* (see Figure 7) and *PKHD1* mutations (Appendix A).


**Association of the Somatic ns Mutation in the TP53 Gene and Dysregulated Pathways in SCC**


The detailed analysis for all the KEGG pathways is presented in Appendix A. Compared to healthy skin tissue, in the SCC samples without the *TP53* ns somatic mutation (n = 3), the genes in the “p53 signaling pathway” were significantly overexpressed by FC 2.21 (95% CI 1.64–2.98), whereas in SCC samples with the *TP53* ns somatic mutation (n = 3), the same pathway genes were somewhat under-expressed by FC −1.32 (95% CI −1.78 to 1.01, see Figure 8). This shows a significant association (interaction *p* = 5.53 × 10^−8^) between *TP53* mutation status and the “p53 signaling pathway”, where the *TP53* mutation is associated with impaired tumor suppression activity.

### 3.6. Gene–Environmental Interaction: Interaction of the Somatic Mutation and Degree of As Exposure on Gene Expression Pathways

All our participants were exposed to As, so to explore the effect of As, we used the UACR at baseline below 192 µg/g creatinine and above 192 µg/g creatinine for comparison. Using the somatic mutation status (no vs. yes) and the baseline UACR level (≤192 µg/g creatinine or low vs. >192 µg/g creatinine or high), we divided the BCC patients into four categories (see Table 4) and compared the expression of the different gene pathways of each group of tissues to the same 16 healthy skin tissues. The overall FC (95% CI) of the pathways in each group are presented in Table 4. The result shows that the presence of the *PTCH1* somatic mutation increases the magnitude of the differential expression of genes in the “basal cell carcinoma pathway” and “hedgehog pathway” in both low and high As exposure groups. It also shows that high As exposure decreases the magnitude of the differential expression in the absence or presence of the *PTCH1* somatic mutation. The differences in the magnitudes of differential expressions were statistically significant, indicated by the interaction *p*-value.

Figure 9 shows the gene expression profiles of the genes in the hedgehog pathway and how their differential expressions are affected by PTCH1 mutation status and the As-exposure level.

In the same way, we also checked the interaction of the *NOTCH1* somatic mutation and As exposure. Table 5 shows how the *NOTCH1* somatic mutation and As exposure status influence the immune response pathways like the “IL-17 signaling pathway”, “Antigen processing and presentation”, and the “p53 signaling pathway”. These results also show a similar trend that the somatic mutation increases the differential expression and high As exposure decreases the magnitude of the differential expression of these pathways.

## 4. Discussion

While UVR exposure and skin sensitivity are known risk factors for NMSC, especially among Caucasians, As exposure through contaminated drinking water may be a major risk factor in other populations. To our knowledge, our current study presents the most comprehensive molecular profiling (more than 400 cancer-related genes) for NMSC in a non-Caucasian population exposed to As. We are unaware of any previous study on NMSC that has considered the fact that apparently healthy, non-lesional human skin exposed to sunlight actually harbors somatic mutations. Our study addressed this fact and identified NMSC-associated somatic mutations that are not found in healthy skin tissue. We acknowledge the weakness of the small sample size and the fact that it would have been ideal if we could sequence normal tumor pairs for all the patients.

A variant seen in a given tissue that is not seen in germline DNA (blood may be used as a proxy) is considered a “somatic mutation”. Because of exposure to UV rays from sunlight, even healthy skin tissue may show a multitude of such somatic mutations resembling a UVR signature. Therefore, unlike many somatic mutations seen in other internal organ cancers, the detection of a somatic mutation in NMSC tissue does not necessarily mean that the detected mutation is a “cancer-associated somatic mutation”. Our study confirms this fact, and by excluding those somatic mutations in healthy skin, we could identify NMSC-associated somatic mutations. Looking at the top 20 genes showing somatic mutations in our study in an As-exposed population from Southeast Asia, we could see that many of the genes are also seen to be mutated in NMSC patients from Caucasian populations worldwide. We did not have patients who were not exposed to As and cannot comment on the cause of mutation or NMSC pathogenesis. Unfortunately, we also did not have any tissue left for measuring As content in the tumor tissue, which could have shed some light if As exposure was associated with NMSC pathogenesis.

Unlike some other cancers, like colorectal or thyroid cancer, where a single-point mutation (like *KRAS* rs#112445441 or *BRAF* V600E) is found in a large proportion of samples, in NMSC, there is no single mutation that is seen in a large number of samples. Rather, sequencing of large genomic regions is needed to detect somatic mutations in a given gene (e.g., *PTCH1* or *NOTCH1*) because the mutations are at different locations in different samples. But, it is interesting to note that, regardless of the difference in position and amino acid change (e.g., c.3583 A>T causing Thr1129Ser change in one sample and c.1313C>A causing Pro504Gln change in another sample), when the samples were grouped together based on BCC-associated *PTCH1* negative or positive non-synonymous somatic mutations, we see a marked difference of the differential expression of many relevant gene pathways. This allows us to utilize these genomic markers for the individualization of targeted therapy if and when they are needed. For example, hedgehog signaling pathway inhibitor small molecules (vismodegib, sonidegib) may be most effective in BCC patients with *PTCH1* mutations who are not exposed to high As, whereas the same therapy may show the least or no response in BCC patients without *PTCH1* somatic mutations exposed to high As (see Table 4 and Figure 8). Currently, both vismodegib and sonidegib are only approved for metastatic or locally advanced BCC [56,57]. But, *PTCH1* mutation status may be used for selecting patients for individualized targeted therapy. In the same line, our data suggest a molecular basis for the potential use of *IL-17* inhibitors in BCC patients with low As exposure with *NOTCH1* somatic mutations (see Table 5). Transcriptomic data were not strongly suggestive of great potential for immune checkpoint inhibitors in these BCC patients; however, they suggested a lower chance of platinum drug resistance in BCC patients with high UACR compared to high platinum drug resistance potential in patients with lower UACR [32].

In a study utilizing ultra-deep sequencing of 74 cancer genes from skin biopsies of normal skin across 234 biopsies of sun-exposed eyelid epidermis from four individuals, Martincorena et al. looked for somatic mutations [58]. The burden of somatic mutations averaged two to six mutations per megabase per cell, similar to that seen in many cancers, and exhibited characteristic signatures of UVR exposure. There was a predominance of C>T mutations and high rates of CC>TT dinucleotide substitutions. *NOTCH1* was the most frequently mutated gene, and 20% of normal skin cells carried a driver mutation in *NOTCH1* [58]. In SCC of the skin and other organs, both copies of *NOTCH1* are frequently inactivated, typically through point mutation combined with copy number alteration. Other frequently-mutated genes include *RBM10*, *FGFR3*, *CDKN2A*, and *NOTCH2* [58].

We found few studies where investigators used fresh-frozen BCC tissue to look at the somatic mutations. In one study, fresh-frozen BCC tumor tissues were obtained from 191 patients, and corresponding normal-appearing skin was available from 115 patients [59]. PCR and Sanger sequencing were performed, and they detected 137 *PTCH1* mutations in 105 tumors with some loss of heterozygosity. For *TP53*, 31% of BCC carried mutations, mostly of the missense type. *TERT* and *DPH3* promoter mutations were present in 113 and 73 cases, respectively. Gene expression analysis found statistically significant higher *TERT* mRNA levels in BCC tumors with *TERT* promoter mutations compared to the tumors without mutations (*p* < 0.001) [59].

In another study, whole genome exome sequencing was performed on a total of 27 pairs of tissue (tumor and normal adjacent healthy skin) [60]. They identified 84,571 cancer sample-specific somatic mutations, of which 42,380 (50.1%) were located in protein-coding regions, and the remaining 42,191 (49.9%) were located in non-coding regions. They showed the relation between the different pathways and mutations, like hedgehog pathways (*PTCH1*, *GL12*, *SMO*), MYCN regulation genes (*MYCN*, *MTOR*, *DYRk3*, *AMBRA1*), filaggrin genes (*FLG*, *FLG2*), and NOTCH genes (*NOTCH1*, *NOTCH2*, *NOTCH3*). They also detected mutations in the non-coding region of *BAD*, *DHODH*, *SPHK2*, *CHCHD2* (also known as *MNRR1*), and *RPS27*. Promoter mutations of *TERT* and *DPH3* were also detected. Mutations were also found in *TP53*, *PTPRD*, *LATS1*, and *ARIDIA* [60]. They found mutations in *TNFAIP2*, which encodes a multifunctional protein playing a role in angiogenesis, inflammation, cell migration and invasion, cytoskeleton remodeling, and cell membrane protrusion formation. In the coding region, they also detected somatic mutations in *EZH2* and *KNSTRN* [60]. Whole-exome sequencing of secondary tumors arising from nevus sebaceous revealed additional genomic alterations in addition to RAS mutations [61].

In SCC, one study found highly mutated *PDE4DIP*, *SYNE1*, and *NOTCH1* genes [62]. They analyzed somatic mutations in SCC in smokers and non-smokers. Mutations of the *ATM*, *RNF213*, *DST*, *RET*, *CYP2C19*, *PKHD1*, *PTPRD*, *SETD2*, *ATR*, *CDKN2A*, *TP53*, *KAT6B*, *FGFR3*, *NOTCH2*, and *NOTCH4* genes, which were common to SCC, were foundin their study [62].

The relation of *PTCH1* mutations and mRNA expression was studied in twenty cases of nevoid basal cell carcinoma (Gorlin) syndrome, an autosomal dominant disorder, using cancer tissue, surrounding healthy tissue, blood DNA, and skin tissue from four healthy people. They detected twelve genomic and five somatic mutations of the *PTCH1* gene. Quantitative PCR was used to determine the mRNA expression levels of *PTCH1*, *SMO*, *GLI3*, and *CCND1* genes in relation to the *PTCH1* mutation. The mRNA expression was highest in BCC tissue, followed by surrounding healthy tissue and the skin tissue of healthy people [63]. They also showed the effect of *PTCH1* mutations on gene expression. In surrounding healthy tissue with *PTCH1* mutations, the mRNA expression was lower for *PHCH1* and *GLI3* genes. On the other hand, they found higher *SMO* and *CCND1* mRNA expressions in the same group. BCC tumors with germline and somatic mutations of *PTCH1* expression levels of *PTCH1*, *SMO*, and *GLI3* were higher compared to those with germline mutations only, but *CCND1* levels were lower in that group [63].

The list of somatic mutations detected in BCC and SCC depends on the number of target genes sequenced in a particular study, the use of whole exome sequencing or whole genome sequencing, and the strictness of criteria for the detection of a somatic mutation. However, some of the most frequently mutated genes are common among the published studies. In that respect, our current study confirms the findings of many of the past studies and also detects some new mutations. We report the NMSC-associated somatic mutations after excluding the somatic mutations seen in the healthy skin tissue, and this study was performed in a Bangladeshi population exposed to As. We acknowledge the fact that we did not perform ultra-deep sequencing to capture rare variants, so we might have missed very rare variants (below 2% frequency). On the other hand, the somatic mutations detected in our study had reasonably high frequency, giving us confidence that the reported mutations are real. Importantly, the associations of the non-synonymous mutations within the frequently mutated genes with the differential expression of genes and major gene pathways further underscore the importance of the findings.

Surgical excision of the NMSC is the first line of management for most of the cases. However, for some recurrent or locally advanced or metastatic cases, targeted therapy may be considered. Keeping that in mind, we analyzed the molecular genomic data in a manner that helps understand the pathogenesis and the utilization of the mutation data for the potential selection of patients for some targeted therapies in the future. We acknowledge that the small sample size did not allow us to test such associations for low-frequency mutations and gene pathways. In the future, we plan to carry out a larger study utilizing the already available biological samples and the clinical follow-up data from the parent BEST study.

## 5. Conclusions

We present the result of somatic mutation detection in NMSC tissue and paired blood samples from participants exposed to As through drinking As-contaminated water and compare them with somatic mutations in healthy skin to identify NMSC-associated mutations. We also show the effect of these mutations on different genes and pathways that may be helpful for targeted therapy in the future.

## Figures and Tables

**Figure 1 cells-13-01056-f001:**
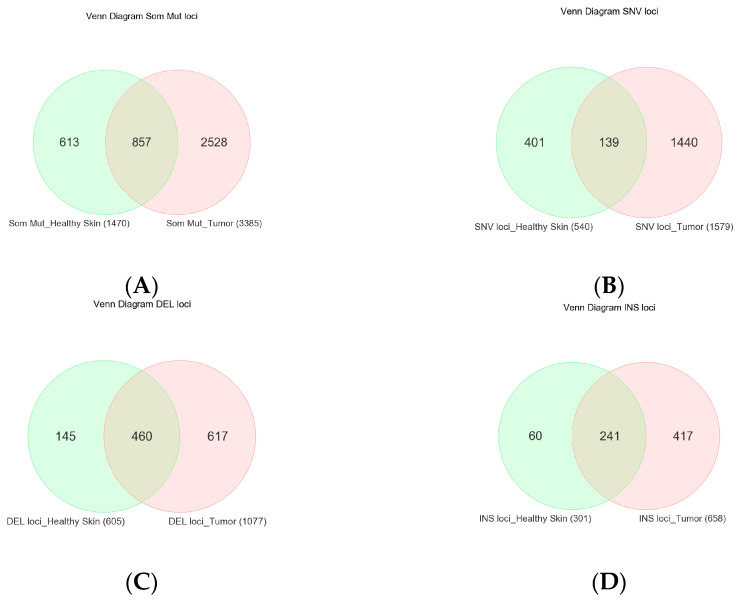
Venn diagram showing the overlap of unique somatic mutation loci among NMSC tissue (in pink) and normal skin tissue (in light green). All types of mutations are shown in the upper left (**A**), SNVs are shown in the upper right (**B**), deletions (DELs) are shown in the lower left (**C**), and insertions (INSs) are shown in the lower right panel (**D**).

**Figure 2 cells-13-01056-f002:**
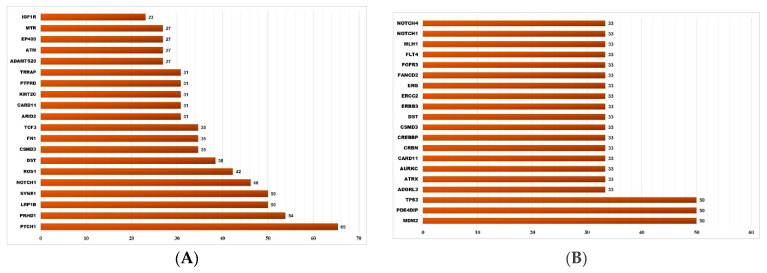
The top 20 genes that had BCC-associated somatic mutations (shown on the left panel (**A**)) and SCC-associated somatic mutations (shown on the right panel (**B**)). The x-axis shows the percentage of samples harboring NMSC-associated somatic mutations in a given gene. The y-axis shows the gene name.

**Figure 3 cells-13-01056-f003:**
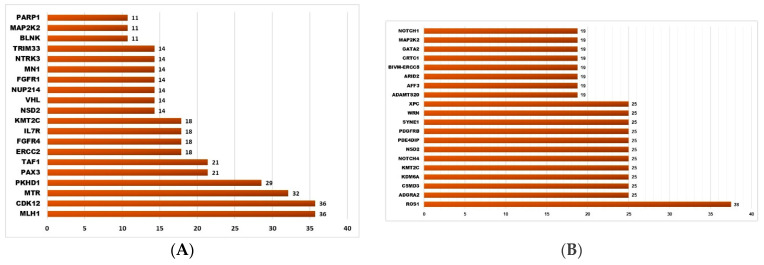
The top 20 genes that had somatic mutations in NMSC and healthy skin (shown on left panel (**A**)) and the top 20 genes that had somatic mutations only in healthy skin tissue (shown on the right panel (**B**)). The x-axis shows the percentage of samples harboring somatic mutations in a given gene. The y-axis shows the gene name.

**Figure 4 cells-13-01056-f004:**
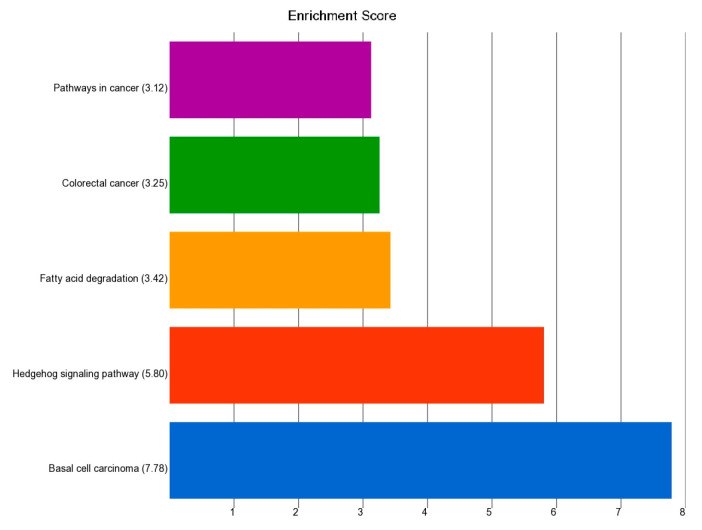
GO enrichment analysis of the top 118 differentially expressed genes (FC ≥ 3 and FDR ≤ 0.05) in BCC tissue compared to healthy skin tissue. The x-axis represents the enrichment score and the y-axis is the group of genes.

**Figure 5 cells-13-01056-f005:**
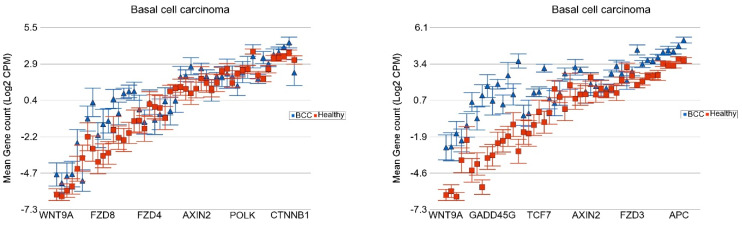
Differential expression of gene pathways in BCC (in blue) compared to healthy skin tissue (in red) by *PTCH1* mutational status. BCC tissues with no non-synonymous somatic mutations in PTCH1 are shown on the left panel and those with mutations are shown on the right panel. Genes are arranged on the x-axis by expression level, and the log2-transformed gene count per million (CPM) is shown on the y-axis. Gene symbols for all the genes could not be shown on the x-axis.

**Figure 6 cells-13-01056-f006:**
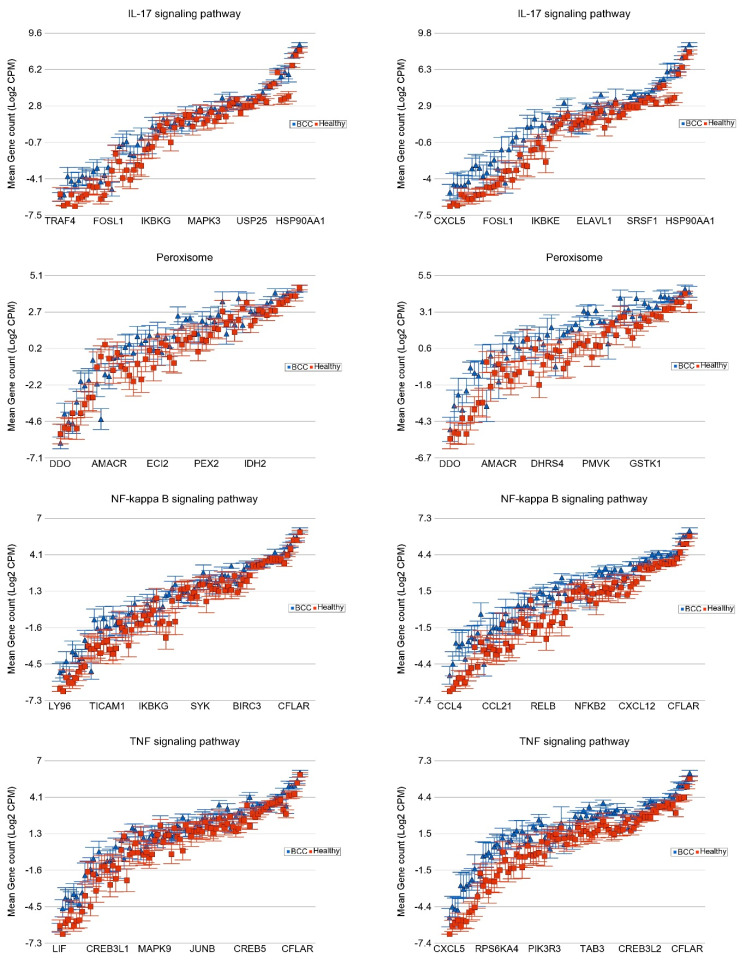
Differential expression of gene pathways in BCC (in blue) compared to healthy skin tissue (in red) by *NOTCH1* mutational status. BCC tissues with no non-synonymous somatic mutations in NOTCH1 are shown on the left panel and those with mutations are shown on the right panel. Genes are arranged on the x-axis by expression level, and the log_2_-transformed gene count per million (CPM) is shown on the y-axis. Gene symbols for all the genes could not be shown on the x-axis.

**Figure 7 cells-13-01056-f007:**
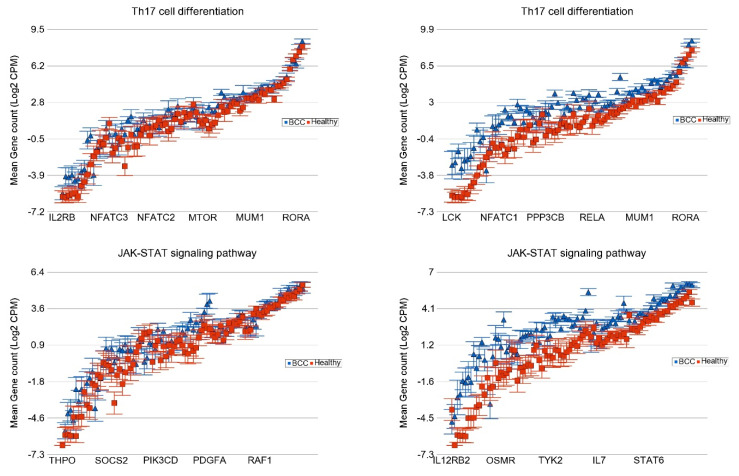
Differential expression of gene pathways in BCC (in blue) compared to healthy skin tissue (in red) by *SYNE1* mutational status. BCC tissues with no non-synonymous somatic mutations in *SYNE1* are shown on the left panel and those with mutations are shown on the right panel. Genes are arranged on the x-axis by expression level, and the log_2_-transformed gene count per million (CPM) is shown on the y-axis. Gene symbols for all the genes could not be shown on the x-axis.

**Figure 8 cells-13-01056-f008:**
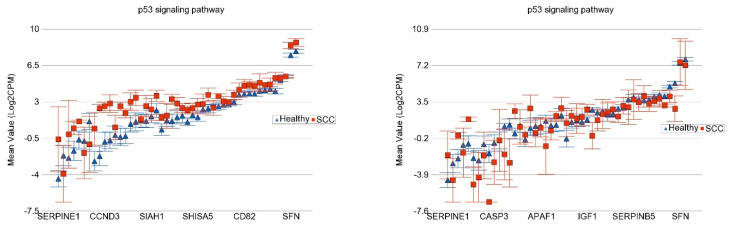
Differential expression of gene pathways in SCC (in red) compared to healthy skin tissue (in blue) by *TP53* mutational status. SCC tissues with no non-synonymous somatic mutations in *TP53* are shown on the left panel and those with mutations are shown on the right panel. Genes are arranged on the x-axis by expression level, and the log_2_-transformed gene count per million (CPM) is shown on the y-axis. Gene symbols for all the genes could not be shown on the x-axis.

**Figure 9 cells-13-01056-f009:**
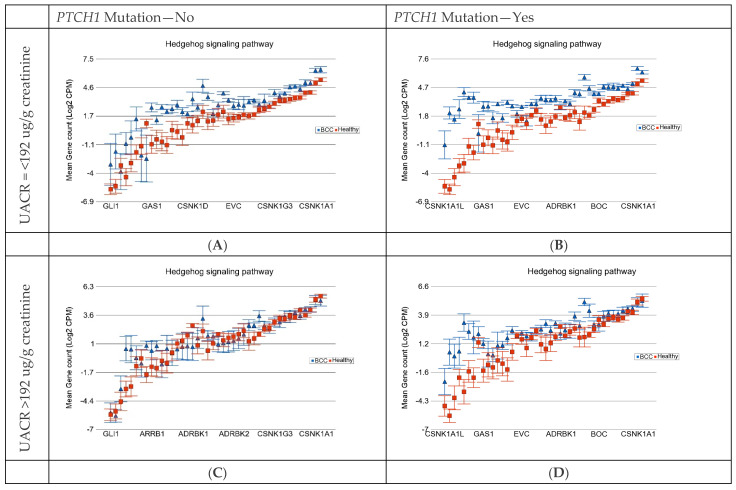
Differential gene expression of hedgehog signaling pathway genes in BCC tissue (in blue) compared to healthy skin tissue (in red). BCC tissues with no somatic mutations in *PTCH1* and low As exposure are shown on the left upper plot (**A**). BCC tissues with somatic mutations in *PTCH1* and low As exposure are shown on the right upper plot (**B**). BCC tissues with no somatic mutations in *PTCH1* and high As exposure are shown on the left lower plot (**C**). BCC tissues with somatic mutations in *PTCH1* and high As exposure are shown on the right lower plot (**D**). Genes are arranged on the x-axis by expression level, and the log_2_-transformed gene count per million (CPM) is shown on the y-axis. Gene symbols for all the genes could not be shown on the x-axis.

**Table 1 cells-13-01056-t001:** Some of the variant metrics by somatic variant type and tissue type.

Somatic Variant Type	Variant Metrics	Median (BCC)	Median (SCC)	Median (Healthy Skin)
All Somatic Variants	Count	10.00	44.00	15.00
	Coverage at the variant loci	169.00	772.50	245.00
	Frequency of the variant	4.74	6.15	4.71
	Q-score of the variant allele	37.00	36.72	36.96
	QUAL of the variant call	200.00	200.00	200.00
SNVs	Count	14.00	357.50	22.00
	Coverage at the variant loci	114.00	974.50	154.00
	Frequency of the variant	8.33	43.89	17.78
	Q-score of the variant allele	37.00	36.86	37.00
	QUAL of the variant call	200.00	200.00	200.00
Insertion	Count	9.00	37.50	15.00
	Coverage at the variant loci	201.00	795.50	317.00
	Frequency of the variant	4.12	4.84	4.63
	Q-score of the variant allele	37.00	36.75	36.93
	QUAL of the variant call	200.00	200.00	200.00
Deletion	Count	8.00	26.00	13.00
	Coverage at the variant loci	189.00	616.00	297.00
	Frequency of the variant	4.17	4.07	3.91
	Q-score of the variant allele	37.00	36.32	36.69
	QUAL of the variant call	200.00	200.00	200.00

**Table 2 cells-13-01056-t002:** The top 118 differentially expressed genes in BCC tissue compared to healthy skin tissue by at least an FC of 3 at an FDR level of 0.05. The comparison of FCs (95% CI) in BCC tissue without a *PTCH1* somatic mutation and BCC tissue with a *PTCH1* mutation status are shown. The genes are arranged in the same order as in Appendix A, where the genes are arranged in ascending order of their *p*-value for the combined analysis. The significant interactions are shown in red.

	BCC without *PTCH1* Mutation		BCC with *PTCH1* Mutation		
Gene Symbol	Fold Change	(95% CI)	Fold Change	(95% CI)	Interaction
					*p*
*CHGA*	37.3	(8.04–172.69)	296.6	(59.86–1469.36)	0.0150
*MPPED1*	35.6	(8.13–156.0)	51.3	(10.99–239.68)	0.6443
*CHCHD7*	3.0	(1.73–5.17)	7.4	(4.18–13.12)	0.0035
*LGR5*	20.5	(4.81–87.16)	84.4	(18.63–382.35)	0.0734
*ABI3BP*	6.3	(2.39–16.80)	27.3	(9.88–75.45)	0.0075
*IRS4*	16.4	(3.67–73.60)	108.6	(22.71–518.88)	0.0229
*LEF1*	9.6	(3.02–30.61)	20.4	(6.1–68.31)	0.2286
*RP11-368P15.3*	15.4	(3.83–62.07)	36.6	(8.55–156.32)	0.2506
*DUSP10*	12.0	(3.05–47.20)	41.5	(9.96–173.2)	0.0960
*PALM*	11.1	(2.83–43.27)	48.6	(11.71–201.7)	0.0480
*DIO2*	3.1	(1.57–6.05)	8.4	(4.14–16.91)	0.0083
*VCAN*	5.4	(1.94–15.17)	26.2	(8.95–76.53)	0.0064
*RGCC*	25.0	(5.30–117.63)	30.5	(6.05–153.7)	0.8094
*LRRCC1*	2.4	(1.40–4.01)	5.1	(2.95–8.84)	0.0090
*SOX4*	2.4	(1.39–3.97)	5.0	(2.88–8.58)	0.0105
*SETBP1*	2.0	(1.26–3.05)	4.9	(3.1–7.82)	0.0003
*DCP1B*	17.0	(3.81–75.70)	35.8	(7.54–170.36)	0.3530
*RP11-157G21.2*	12.8	(3.57–45.78)	16.4	(4.34–62.12)	0.7144
*NNMT*	15.5	(3.28–73.10)	53.8	(10.66–271.7)	0.1390
*TGS1*	2.8	(1.61–4.86)	3.6	(2.04–6.44)	0.3826
*SPON2*	4.1	(1.60–10.47)	23.4	(8.79–62.22)	0.0012
*PROCR*	11.1	(2.71–45.07)	41.0	(9.45–177.31)	0.0875
*LPL*	12.4	(2.97–51.52)	35.6	(8.04–157.49)	0.1714
*BNC2*	2.5	(1.34–4.77)	9.5	(4.9–18.47)	0.0003
*SYNJ1*	7.1	(2.21–22.57)	21.2	(6.3–71.02)	0.0833
*TM4SF1*	5.0	(1.82–13.9)	18.7	(6.49–54.02)	0.0196
*PTCH1*	2.8	(1.47–5.33)	6.0	(3.07–11.77)	0.0311
*MMP11*	11.4	(2.45–53.18)	77.3	(15.54–384.35)	0.0245
*MEX3A*	3.7	(1.48–9.08)	17.6	(6.86–45.33)	0.0023
*RP11-433O3.1*	16.0	(3.03–84.15)	60.4	(10.68–341.94)	0.1400
*TUBA1A*	3.2	(1.53–6.52)	6.6	(3.11–14.06)	0.0622
*NXN*	2.8	(1.51–5.11)	4.3	(2.29–8.13)	0.1822
*TMCO3*	10.3	(2.53–42.01)	30.9	(7.14–133.59)	0.1497
*PYCR2*	14.3	(2.87–71.24)	51.0	(9.55–272.2)	0.1444
*ALCAM*	2.7	(1.42–5.22)	5.6	(2.85–11.02)	0.0433
*RP11-74M13.4*	−13.6	(−56.71–3.24)	−20.3	(-90.17–4.55)	0.6003
*MARCKSL1*	9.8	(2.15–44.09)	59.8	(12.38–288.3)	0.0294
*SCAMP5*	8.2	(2.07–32.72)	38.7	(9.18–163.16)	0.0412
*VASH2*	8.2	(2.06–32.65)	38.9	(9.21–164.60)	0.0403
*HDGFRP3*	9.6	(2.24–41.29)	40.4	(8.83–184.54)	0.0715
*KRT17*	4.7	(1.74–12.72)	12.2	(4.31–34.25)	0.0802
*DCLRE1A*	9.4	(2.30–38.67)	31.3	(7.19–136.32)	0.1176
*ATP5F1*	4.7	(1.81–12.24)	9.7	(3.56–26.24)	0.1641
*RP11-366L20.2*	11.2	(2.50–50.45)	36.1	(7.53–172.85)	0.1519
*BASP1*	2.1	(1.20–3.61)	5.8	(3.24–10.23)	0.0012
*DSC2*	3.2	(1.4–7.29)	10.7	(4.5–25.24)	0.0091
*IGF2BP2*	2.2	(1.21–3.94)	6.3	(3.39–11.55)	0.0017
*FBN3*	8.1	(1.67–39.62)	151.1	(28.98–788.07)	0.0013
*CSPG4*	7.1	(1.83–27.53)	39.1	(9.51–161.04)	0.0228
*CCDC152*	5.2	(1.77–15.01)	13.8	(4.54–42.13)	0.0902
*IGF2BP1*	−12.0	(−48.57–2.94)	−17.7	(−76.48–4.1)	0.6008
*SOBP*	6.5	(1.99–21.18)	16.0	(4.67–54.9)	0.1587
*TMEM176B*	9.5	(2.11–42.44)	45.4	(9.48–217.22)	0.0565
*RP11-301G23.1*	8.8	(1.97–39.52)	52.7	(11.03–251.33)	0.0307
*ACAA2*	11.5	(2.30–57.16)	53.7	(10.05–287.19)	0.0781
*SHOX2*	4.7	(1.53–14.21)	22.7	(7.12–72.64)	0.0108
*COA7*	21.1	(5.65–78.70)	7.1	(1.79–27.97)	0.1268
*RUNX1T1*	3.6	(1.43–8.85)	12.4	(4.78–32.02)	0.0139
*ITFG3*	3.2	(1.65–6.35)	3.9	(1.94–7.9)	0.6011
*GRM5*	−9.3	(−34.5–2.49)	−16.0	(−62.86–4.05)	0.4416
*PLAG1*	9.6	(2.26–41.08)	29.8	(6.57–135.5)	0.1504
*SKIV2L*	12.8	(3.14–52.23)	15.2	(3.5–65.88)	0.8202
*HMGN1P38*	5.5	(2.1–14.27)	6.5	(2.41–17.76)	0.7300
*CTB-167B5.2*	9.9	(2.25–43.22)	31.5	(6.75–147.23)	0.1471
*VPS37D*	6.1	(1.71–21.50)	27.5	(7.35–103.18)	0.0302
*OSCP1*	13.7	(2.88–64.61)	25.9	(5.12–131.33)	0.4421
*C1QTNF1*	19.1	(3.73–97.99)	22.7	(4.13–124.97)	0.8438
*ACADL*	−29.6	(−148.16–5.92)	−13.3	(−71.41–2.48)	0.3556
*GJB6*	5.5	(1.93–15.78)	9.7	(3.24–29.02)	0.3184
*PTCH2*	9.5	(1.8–49.61)	101.3	(17.97–570.53)	0.0104
*RAB28*	12.7	(2.75–58.71)	24.4	(4.94–120.48)	0.4273
*CASC14*	12.1	(2.4–60.91)	39.6	(7.33–213.81)	0.1754
*LTBP1*	2.9	(1.25–6.59)	12.1	(5.09–28.84)	0.0023
*ALYREF*	13.3	(3.0–58.87)	18.0	(3.81–85.04)	0.7035
*LITAF*	4.9	(1.91–12.46)	6.6	(2.46–17.44)	0.5564
*TMEM104*	6.2	(1.54–25.25)	50.2	(11.68–215.66)	0.0078
*NUDCD1*	12.4	(2.95–52.09)	15.2	(3.39–67.72)	0.7944
*SLCO2A1*	13.7	(2.77–67.39)	26.5	(5.02–140.11)	0.4391
*SH3BP4*	5.3	(1.92–14.42)	7.8	(2.71–22.29)	0.4691
*RP4-765C7.2*	12.0	(2.29–63.0)	43.8	(7.77–246.52)	0.1502
*RP11-5N11.5*	5.2	(1.29–20.68)	87.7	(20.63–372.88)	0.0004
*FMO2*	6.4	(1.95–20.71)	12.3	(3.59–42.36)	0.2968
*AC005013.1*	5.8	(1.82–18.61)	12.8	(3.79–42.87)	0.2120
*AC187652.1*	11.9	(2.99–47.46)	11.8	(2.78–49.71)	0.9851
*GALNT15*	9.6	(2.41–38.38)	15.6	(3.67–65.88)	0.5174
*EDN1*	5.1	(1.77–14.59)	9.5	(3.16–28.55)	0.2714
*CREB5*	2.3	(1.13–4.65)	8.7	(4.15–18.04)	0.0010
*RP11-73E6.2*	−8.1	(−28.01–2.33)	−10.7	(−39.29–2.93)	0.6692
*TBX1*	6.3	(1.53–25.63)	39.1	(9.01–169.72)	0.0190
*RAC3*	13.1	(3.15–54.56)	11.9	(2.69–52.64)	0.8978
*RP3-512B11.3*	9.3	(2.06–41.58)	27.4	(5.71–131.35)	0.1821
*C1QA*	11.8	(2.11–66.46)	52.4	(8.66–316.86)	0.1128
*COA3*	13.0	(2.8–60.32)	18.3	(3.69–90.76)	0.6775
*RP11-384C21.9*	−7.9	(−31.63–1.98)	−20.2	(−85.61–4.76)	0.2112
*TGFB2*	8.3	(2.22–31.06)	13.6	(3.42–53.58)	0.4900
*STMN1*	2.6	(1.19–5.42)	8.0	(3.61–17.48)	0.0072
*GLI2*	7.5	(1.86–30.38)	22.6	(5.26–96.75)	0.1466
*UBAC2*	7.2	(2.09–24.74)	11.3	(3.11–40.92)	0.4971
*SOX18*	10.9	(1.99–59.42)	48.9	(8.33–286.95)	0.1039
*IFT88*	5.4	(1.86–15.56)	8.0	(2.65–24.26)	0.4840
*RP11-496I9.1*	11.6	(2.19–60.82)	35.5	(6.27–201.05)	0.2107
*RN7SL776P*	−13.7	(−57.39–3.25)	−10.8	(−48.48–2.42)	0.7635
*FSCN1*	7.0	(1.96–25.13)	13.4	(3.542–50.60)	0.3471
*SLC7A2*	3.8	(1.37–10.49)	12.3	(4.24–35.44)	0.0361
*RBP1*	10.9	(2.52–46.79)	15.4	(3.36–70.83)	0.6533
*TMEM217*	−10.5	(−45.1–2.45)	−15.7	(−71.82–3.44)	0.6061
*LRIG3*	4.7	(1.6–13.51)	10.1	(3.31–30.56)	0.1816
*PELI2*	4.1	(1.44–11.52)	11.5	(3.89–34.03)	0.0676
*TRIL*	8.3	(1.98–34.59)	20.2	(4.55–89.76)	0.2476
*ARPC1B*	3.8	(1.52–9.69)	7.3	(2.78–19.23)	0.1974
*CAV1*	2.2	(1.13–4.06)	5.7	(2.94–11.13)	0.0062
*GPX8*	7.5	(1.92–29.08)	17.1	(4.15–70.41)	0.2594
*INPP4A*	5.2	(1.75–15.31)	9.0	(2.92–28.01)	0.3391
*ADAM23*	8.3	(1.84–36.98)	27.6	(5.79–131.96)	0.1376
*PNMA1*	9.1	(1.93–43.07)	29.3	(5.8–148.32)	0.1646
*MGAT5*	3.7	(1.50–8.99)	6.6	(2.6–16.79)	0.2251
*IRX1*	5.4	(1.22–23.96)	74.3	(15.73–351.2)	0.0020
*UST*	8.4	(1.92–36.55)	22.6	(4.86–104.94)	0.2130

**Table 3 cells-13-01056-t003:** BCC-associated non-synonymous somatic mutations in the *PTCH1* gene. The coding region changes and the amino acid changes are reported in multiple sources. An “*” in the amino acid change column indicates a translation termination codon.

Coordinate	Reference Allele	Variant Allele	Exact Match	Coding Region Change	Amino Acid Change
chr9:98211572	T	A	clinvar_hg19	NM_000264.3:c.3583A>T; NM_001083602.1:c.3385A>T; NM_001083603.1:c.3580A>T; NM_001083604.1:c.3130A>T; NM_001083605.1:c.3130A>T; NM_001083606.1:c.3130A>T; NM_001083607.1:c.3130A>T	NP_000255.2:p.Thr1195Ser; NP_001077071.1:p.Thr1129Ser; NP_001077072.1:p.Thr1194Ser; NP_001077073.1:p.Thr1044Ser; NP_001077074.1:p.Thr1044Ser; NP_001077075.1:p.Thr1044Ser; NP_001077076.1:p.Thr1044Ser
chr9:98239132	G	T	clinvar_hg19	NM_000264.3:c.1511C>A; NM_001083602.1:c.1313C>A; NM_001083603.1:c.1508C>A; NM_001083604.1:c.1058C>A; NM_001083605.1:c.1058C>A; NM_001083606.1:c.1058C>A; NM_001083607.1:c.1058C>A	NP_000255.2:p.Pro504Gln; NP_001077071.1:p.Pro438Gln; NP_001077072.1:p.Pro503Gln; NP_001077073.1:p.Pro353Gln; NP_001077074.1:p.Pro353Gln; NP_001077075.1:p.Pro353Gln; NP_001077076.1:p.Pro353Gln
chr9:98242779	C	A		NM_000264.3:c.838G>T; NM_001083602.1:c.640G>T; NM_001083603.1:c.835G>T; NM_001083604.1:c.385G>T; NM_001083605.1:c.385G>T; NM_001083606.1:c.385G>T; NM_001083607.1:c.385G>T	NP_000255.2:p.Glu280*; NP_001077071.1:p.Glu214*; NP_001077072.1:p.Glu279*; NP_001077073.1:p.Glu129*; NP_001077074.1:p.Glu129*; NP_001077075.1:p.Glu129*; NP_001077076.1:p.Glu129*
chr9:98242797	G	A		NM_000264.3:c.820C>T; NM_001083602.1:c.622C>T; NM_001083603.1:c.817C>T; NM_001083604.1:c.367C>T; NM_001083605.1:c.367C>T; NM_001083606.1:c.367C>T; NM_001083607.1:c.367C>T	NP_000255.2:p.Gln274*; NP_001077071.1:p.Gln208*; NP_001077072.1:p.Gln273*; NP_001077073.1:p.Gln123*; NP_001077074.1:p.Gln123*; NP_001077075.1:p.Gln123*; NP_001077076.1:p.Gln123*
chr9:98248001	G	A		NM_000264.3:c.550C>T; NM_001083602.1:c.352C>T; NM_001083603.1:c.547C>T; NM_001083604.1:c.97C>T; NM_001083605.1:c.97C>T; NM_001083606.1:c.97C>T; NM_001083607.1:c.97C>T	NP_000255.2:p.Gln184*; NP_001077071.1:p.Gln118*; NP_001077072.1:p.Gln183*; NP_001077073.1:p.Gln33*; NP_001077074.1:p.Gln33*; NP_001077075.1:p.Gln33*; NP_001077076.1:p.Gln33*
chr9:98244299	A	T		NM_000264.3:c.678T>A; NM_001083602.1:c.480T>A; NM_001083603.1:c.675T>A; NM_001083604.1:c.225T>A; NM_001083605.1:c.225T>A; NM_001083606.1:c.225T>A; NM_001083607.1:c.225T>A	NP_000255.2:p.Cys226*; NP_001077071.1:p.Cys160*; NP_001077072.1:p.Cys225*; NP_001077073.1:p.Cys75*; NP_001077074.1:p.Cys75*; NP_001077075.1:p.Cys75*; NP_001077076.1:p.Cys75*
chr9:98268818	T	A		NM_000264.3:c.265A>T; NM_001083602.1:c.67A>T; NM_001083603.1:c.262A>T; NM_001083604.1:c.-189A>T; NM_001083605.1:c.-189A>T; NM_001083606.1:c.-189A>T; NM_001083607.1:c.-189A>T	NP_000255.2:p.Lys89*; NP_001077071.1:p.Lys23*; NP_001077072.1:p.Lys88*
chr9:98241336	C	T	clinvar_hg19	NM_000264.3:c.1161G>A; NM_001083602.1:c.963G>A; NM_001083603.1:c.1158G>A; NM_001083604.1:c.708G>A; NM_001083605.1:c.708G>A; NM_001083606.1:c.708G>A; NM_001083607.1:c.708G>A	NP_000255.2:p.Trp387*; NP_001077071.1:p.Trp321*; NP_001077072.1:p.Trp386*; NP_001077073.1:p.Trp236*; NP_001077074.1:p.Trp236*; NP_001077075.1:p.Trp236*; NP_001077076.1:p.Trp236*
chr9:98268719	C	A		NM_000264.3:c.364G>T; NM_001083602.1:c.166G>T; NM_001083603.1:c.361G>T; NM_001083604.1:c.-90G>T; NM_001083605.1:c.-90G>T; NM_001083606.1:c.-90G>T; NM_001083607.1:c.-90G>T	NP_000255.2:p.Glu122*; NP_001077071.1:p.Glu56*; NP_001077072.1:p.Glu121*
chr9:98215785	C	A		NM_000264.3:c.3424G>T; NM_001083602.1:c.3226G>T; NM_001083603.1:c.3421G>T; NM_001083604.1:c.2971G>T; NM_001083605.1:c.2971G>T; NM_001083606.1:c.2971G>T; NM_001083607.1:c.2971G>T	NP_000255.2:p.Gly1142*; NP_001077071.1:p.Gly1076*; NP_001077072.1:p.Gly1141*; NP_001077073.1:p.Gly991*; NP_001077074.1:p.Gly991*; NP_001077075.1:p.Gly991*; NP_001077076.1:p.Gly991*
chr9:98218658	C	T		NM_000264.3:c.3206G>A; NM_001083602.1:c.3008G>A; NM_001083603.1:c.3203G>A; NM_001083604.1:c.2753G>A; NM_001083605.1:c.2753G>A; NM_001083606.1:c.2753G>A; NM_001083607.1:c.2753G>A	NP_000255.2:p.Gly1069Asp; NP_001077071.1:p.Gly1003Asp; NP_001077072.1:p.Gly1068Asp; NP_001077073.1:p.Gly918Asp; NP_001077074.1:p.Gly918Asp; NP_001077075.1:p.Gly918Asp; NP_001077076.1:p.Gly918Asp
chr9:98239830	T	A		NM_000264.3:c.1502A>T; NM_001083602.1:c.1304A>T; NM_001083603.1:c.1499A>T; NM_001083604.1:c.1049A>T; NM_001083605.1:c.1049A>T; NM_001083606.1:c.1049A>T; NM_001083607.1:c.1049A>T	NP_000255.2:p.Gln501Leu; NP_001077071.1:p.Gln435Leu; NP_001077072.1:p.Gln500Leu; NP_001077073.1:p.Gln350Leu; NP_001077074.1:p.Gln350Leu; NP_001077075.1:p.Gln350Leu; NP_001077076.1:p.Gln350Leu

**Table 4 cells-13-01056-t004:** Effect of the *PTCH1* somatic mutation and As exposure on the differential expression of gene pathways in BCC. Result from Gene set ANOVA analysis showing the FC (95% CI) of different pathways in BCC samples compared to healthy skin tissue. Patients were divided by *PTCH1* somatic mutation status (no vs. yes) and level of As exposure—baseline UACR (low: ≤192 µg/g creatinine vs. high: >192 µg/g creatinine).

			Basal Cell Carcinoma		Hedgehog Signaling Pathway		Antigen Processing and Presentation	
UACR			*PTCH1* Mut −ve	*PTCH1* Mut +ve	*PTCH1* Mut −ve	*PTCH1* Mut +ve	*PTCH1* Mut −ve	*PTCH1* Mut +ve
	≤192 µg/g	FC	4.05	6.92	2.93	5.58	4.77	6.25
		95% CI	(3.03–5.4)	(5.30–9.04)	(2.23–3.85)	(4.34–7.17)	(3.72–6.09)	(4.98–7.83)
	≥192 µg/g	FC	1.33	2.98	1.37	3.01	1.08	2.05
		95% CI	(1.08–1.65)	(2.35–3.7)	(1.12–1.67)	(2.40–3.76)	(−1.098–1.30)	(1.67–2.50)
Interaction p			1.68 × 10^−31^		3.17 × 10^−24^		2.75 × 10^−52^	

**Table 5 cells-13-01056-t005:** Effect of *NOTCH1* somatic mutations and As-exposure on the differential expression of gene pathways in BCC. Result from Gene set ANOVA analysis showing the FC (95% CI) of different pathways in BCC samples compared to healthy skin tissue. Patients were divided by *NOTCH1* somatic mutation status (no vs. yes) and level of As exposure—baseline UACR (low: ≤192 µg/g creatinine vs. high: >192 µg/g creatinine).

			IL17 Signaling Pathway		Antigen Processing and Presentation		*p53* Signaling Pathway	
UACR			*NOTCH1* Mut −ve	*NOTCH1* Mut +ve	*NOTCH1* Mut −ve	*NOTCH1* Mut +ve	*NOTCH1* Mut −ve	*NOTCH1* Mut +ve
	≤192 µg/g	FC	5.13	6.45	5.44	6.09	4.96	5.77
		95% CI	(4.18–6.31)	(4.93–8.43)	(4.39–6.71)	(4.62–8.02)	(4.04–6.08)	(4.42–7.53)
	≥192 µg/g	FC	1.12	2.29	1.17	2.58	1.26	2.66
		95% CI	(−1.04–1.32)	(1.80–2.91)	(−1.01–1.38)	(2.01–3.30)	(1.07–1.48)	(2.09–3.37)
Interaction p			6.27 × 10^−60^		2.45 × 10^−54^		2.49 × 10^−48^	

## Data Availability

All the supporting data are presented in the tables presented in the main manuscript and Appendix A.

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
