# Peer review of "Molecular Profiling and the Interaction of Somatic Mutations with Transcriptomic Profiles in Non-Melanoma Skin Cancer (NMSC) in a Population Exposed to Arsenic"

_cells, 2024, doi:10.3390/cells13121056_

Round 1
Reviewer 1 Report
Comments and Suggestions for Authors
Farzana Jasmine and colleagues presented a research article aimed at evaluating the molecular features of non-melanoma skin cancers in an arsenic-exposed population. For this purpose, the authors performed an NGS panel to identify somatic mutations affecting key genes involved in tumor progression. Overall, the experimental approach used is very simple and not innovative. In addition, a lot of analyses were not performed (e.g. variant filtrations according to population database, GO and GSEA analyses, etc.). The manuscript is also written in a confusing manner. For all these reasons, the manuscript cannot be accepted for publication unless extensive revisions. Please see and carefully address the following minor/major revisions:
1) In line 49-50 please use the expression “phototype” instead of “pigment status”. Same comment in line 61 (substitute the expression “light skin color” with “clear phototype”);
2) Some parts of the manuscript are not fluently written. English editing is suggested;
3) In the introduction section, please use the full form of UVR the first time mentioned;
4) Before describing the molecular alteration observed in BCC and SCC, please briefly describe the pathogenetic mechanisms mediated by arsenic and responsible for the increased risk of non-melanoma skin cancer;
5) From line 70 to line 81, you described some common mutations observed in BCC and SCC. However, you provided very old references (from 2013 to 2017). Notably, most of metagenomics and transcriptomics studies on tumors have been performed in recent years through the use of high-throughput technologies. Therefore, you are encouraged to provide updated data and updated references.
For this purpose, please see:
- PMID: 32899768
- PMID: 35625975
- PMID: 37760432
6) The second half part of the Introduction section is too verbose and should be replaced in the Discussion section;
The following sentence was repeated to many times: “Thus, among these As-exposed study population, 2.2% of the male and 1.3% of the female participants developed BCC, while 0.4% of the male and 0.2% of the female participants developed SCC over the six-year follow-up-“ Please remove it from the Material and Methods section;
7) It is suggested to divide the Materials and Methods section in different subsections;
8) Did you filter the somatic mutations identified in your samples with polymorphic mutations deposited in population databases (gnomAD, 1000Genome, etc.)? This is fundamental to avoid bias in your results;
9) Gene Set Enrichment and Gene Ontology analyses are strongly recommended to establish the functional role of the somatic variants here identified. Please address this critical issue;
10) Please, avoid expressions like “whites” in the following sentence: While UVR exposure and skin sensitivity are known risk factors for NMSC, especially among whites,....”. Please use the more appropriate term “Caucasian”;
11) A key limitation of the study is the limited number of tumor samples and controls analyzed. You have to clearly claim this limitation in the conclusive remarks;
12) Avoid obvious sentences like “A structural change in DNA in a given tissue that is not seen in germline DNA (blood may be used as proxy) is considered as “somatic mutation”.
Comments on the Quality of English LanguageThe manuscript is not fluently written and needs revisions.
Reviewer 2 Report
Comments and Suggestions for Authors
The authors presented a very interesting manuscript regarding the somatic mutations in non-melanoma skin cancer of people exposed to arsenic. The study was done following 7,000 adults man and women along 6 years, from which group some presented non-melanoma skin cancer. The authors showed the changes in the transcriptomic profile, which is an interesting result to consider since non-melanoma skin cancer is a highly incident disease.
1 – In the abstract I would suggest to write what is BCC and SCC (lines 20-21).
2 – The introduction is very well written but maybe would be interesting to develop more the effects of arsenic to populations exposed to this carcinogenic substance (lines 64-66).
3 – In figure 8, the figure "line 2 side right, the graph shows “tissue” written, while all the other figures do not. Please take a standard for all the legends from the graphs.
4 – Take attention to lines 480 and 481, it seems that it is written in another font of text.
Round 2
Reviewer 1 Report
Comments and Suggestions for Authors
The authors significantly improved their manuscript and well-addressed my previous comments.